# Age and Comorbidities as Risk Factors for Severe COVID-19 in Mexico, before, during and after Massive Vaccination

**DOI:** 10.3390/vaccines11111676

**Published:** 2023-11-02

**Authors:** Lenin Domínguez-Ramírez, Francisca Sosa-Jurado, Guadalupe Díaz-Sampayo, Itzel Solis-Tejeda, Francisco Rodríguez-Pérez, Rosana Pelayo, Gerardo Santos-López, Paulina Cortes-Hernandez

**Affiliations:** 1Population Health and Metadynamics Lab, Centro de Investigacion Biomedica de Oriente, Instituto Mexicano del Seguro Social Atlixco, Atlixco 74360, Puebla, Mexico; j.lenin.dominguez.ramirez@gmail.com (L.D.-R.); guadalupe.diaz.sampayo@gmail.com (G.D.-S.);; 2Virology Lab, Centro de Investigacion Biomedica de Oriente, Instituto Mexicano del Seguro Social, Atlixco 74360, Puebla, Mexico; sosajurado@hotmail.com (F.S.-J.); gerardo.santos.lopez@gmail.com (G.S.-L.); 3Oncoimmunology and Cytomics Lab, Centro de Investigacion Biomedica de Oriente, Instituto Mexicano del Seguro Social, Atlixco 74360, Puebla, Mexico; rosana.pelayo@imss.gob.mx; 4Unidad de Educación e Investigación, Instituto Mexicano del Seguro Social, Dirección de Prestaciones Médicas, Mexico City 06725, Mexico

**Keywords:** SARS-CoV-2 vaccines, obesity, diabetes, chronic kidney disease, middle-age adults, COVID-19, massive vaccination in Mexico

## Abstract

During 2020–2023, Mexico had a large COVID-19 emergency with >331,000 adult deaths and one of the highest excess mortalities worldwide. Age at COVID-19 death has been lower in Mexico than in high-income countries, presumably because of the young demographics and high prevalence of chronic metabolic diseases in young and middle-aged adults. SARS-CoV-2 vaccination covered 85% of adults with at least one dose and 50% with booster(s) up to April 2022. No new vaccination efforts or updated boosters were introduced until October 2023; thus, we explored the public health impact of massive SARS-CoV-2 vaccination against ancestral strains and asked whether their real-world protection has persisted through time. We compared three periods with respect to vaccine roll-outs: before, during and after vaccine introduction in a national retrospective cohort of >7.5 million COVID-19 cases. The main findings were that after vaccination, COVID-19 mortality decreased, age at COVID-19 death increased by 5–10 years, both in populations with and without comorbidities; obesity stopped being a significant risk factor for COVID-19 death and protection against severe disease persisted for a year after boosters, including at ages 60–79 and 80+. Middle-aged adults had the highest protection from vaccines/hybrid immunity and they more than halved their proportions in COVID-19 deaths.

## 1. Introduction

In the first three years since SARS-CoV-2 arrival in Mexico, six COVID-19 waves swept the country, with gradually decreasing lethality as population immunity grew [1,2,3]. The six waves are analyzed here and comprise the entire COVID-19 public health emergency declared by Mexican authorities from 30 March 2020 to 9 May 2023 (38 months). Massive anti-SARS-CoV-2 vaccination for adults (18+) happened in Mexico during a 15-month period during the emergency, from February 2021 to April 2022, with vaccines from seven developers (Figure A1 in Appendix B) to maximize coverage and 85% of adults received at least one dose. Less than 1/3 of adult doses corresponded to mRNA vaccines (Figure A1), rendering the Mexican strategy different from that of higher-income countries, in particular the USA, that used a majority of mRNA vaccines. In Mexico, SARS-CoV-2 vaccination was free and voluntary with no mandates in place for employment or education and with generally good acceptance by adults [1].

After their primary doses in 2021, Mexican adults 18–59 yo (years old) had access to one vaccine booster, while the 60+, health personnel and adults with chronic comorbidities could receive up to two boosters for a maximum of four doses [1]. Booster doses were given from December 2021 to April 2022. Around 80% of boosters were with adenoviral vector vaccines: 59% with ChAdOx1-S/nCoV-19 AZD1222 (University of Oxford, UK/Astra Zeneca), 10% with Ad5-nCoV (Cansino Biologics, Tianjin, China), 10% with Gam-COVID-Vac/Sputnik V (Gamaleya Research Institute, Moscow, Russia); while the remaining 20% were with BNT162b2 (Pfizer-BioNTech) mRNA vaccines, all directed against ancestral strains [1]. Many boosters were heterologous with respect to the primary series, although no direct quantification of the combinations used has been published. No updated anti-SARS-CoV-2 vaccines targeting Omicron variants have been introduced in Mexico and most adults received their last booster by April 2022 when coverage reached 85% of adults with at least one dose and almost 50% with a complete vaccine series and one or two boosters (Figure A1). Children in Mexico achieved lower coverage in the period of analysis, as vaccination has not been available to those aged 0–4 yo (8.2% of the population). Children 5–17 yo that comprise 22% of the population, were eligible for two pediatric doses of the BNT162b2 Pfizer-BioNtech mRNA vaccine in 2022, the only COVID-19 vaccine with emergency authorization for children in Mexico, with no access yet to boosters or third doses (Figure A1).

SARS-CoV-2 variants continued to circulate in Mexico in the year following adult vaccination (May 2022–June 2023), mostly characterized by Omicron sub-lineages BA.2, BA.5, BQ.1, BW.1 and XBB.1 [4], producing COVID-19 waves 5 and 6. Between August and November 2022, more than 94% of the population had anti-N SARS-CoV-2 antibodies [1], suggesting extensive virus circulation and high levels of hybrid immunity, from vaccination and infection in all age groups. As of September 2023, no additional booster strategies are in place for Mexican adults who completed vaccination by April 2022, and it is still unknown whether periodic vaccinations will be needed to maintain population protection against severe COVID-19 or how should these be organized now that the public emergency has passed. Surveillance is crucial to detect if immunity has decreased with time since vaccination. In particular, the surveillance of vulnerable groups, like the multimorbid and the elderly could aid in detecting waning COVID-19 immunity.

The Mexican population has over 90 million adults 18+, with a high proportion (40%) of middle-aged 40–64 yo [5], that were severely affected in the initial COVID-19 waves, accumulating 50% of hospitalizations and 45% of deaths before vaccination [3]. Additionally, Mexico has a world-leading prevalence of overweight and obesity, present in >75% of adults, which favors the early development of metabolic diseases, resulting in almost 30% of adults 40+ being hypertense and 20% being diabetic [6]. Many early studies in Mexico correlated comorbidities with the risk of severe COVID-19, facilitated by the national COVID-19 notification system [7], that during the entire emergency interrogated the presence of nine comorbidities in every suspected case: obesity, diabetes, hypertension, cardiovascular disease (CVD), chronic kidney disease (CKD), chronic obstructive pulmonary disease (COPD), immunosuppression (IS), asthma and smoking, plus the option “other comorbidity(ies)”. In early Mexican studies, it was shown that comorbidities, in particular obesity, increased the risk of severe COVID-19 in young and middle-aged adults [8], as described in other countries [9]. There are few analyses on how comorbidities behaved as risk factors for severe COVID-19 after massive vaccination. Here, we first compiled and ranked the published risk factors for severe COVID-19 in Mexico, from studies before vaccination. Then we analyzed the frequency of comorbidities in COVID-19 deaths, their case fatality rate and risk, before, during and after vaccine introduction, to explore how the main risk factors for COVID-19 fatality changed nationwide with massive vaccination. We report that obesity decreased importantly as a risk factor in adult COVID-19 fatalities as population immunity increased.

## 2. Materials and Methods

### 2.1. Bibliographic Review of Risk Factors Reported before Vaccination

A search and review of full-text published manuscripts in English or Spanish was conducted, with keywords SARS-CoV-2, COVID-19 risk factor, COVID-19 mortality, Mexico. We included 23 studies published from February 2020 to December 2021 (Appendix A) [8,10,11,12,13,14,15,16,17,18,19,20,21,22,23,24,25,26,27,28,29,30,31], with the following criteria: studies in the Mexican population, before COVID-19 vaccine rollouts, which analyzed risk factors for presenting COVID-19 or for developing serious illness that required hospitalization, assisted mechanical ventilation (AMV) or intensive care unit (ICU), and studies that analyzed COVID-19 mortality and/or lethality (case fatality rate, CFR). Most studies analyzed data from the same national dataset [7] that regularly updates the characteristics of each COVID-19 case in Mexico, not including asymptomatic SARS-CoV-2 infections. (Data for the entire sanitary emergency from this platform was analyzed in the second part of this manuscript).

All cases in the published studies were confirmed by RT-PCR and occurred in unvaccinated individuals, during the first two COVID-19 waves in Mexico, with data up to January 2021. The authors used risk measures or adjustment models to associate the presence of comorbidity with hospitalization, AMV, ICU admission, or pneumonia. Some authors [10,11,12] validated the Mexican population’s pre-existing scores.

### 2.2. Analysis of the COVID-19 Sanitary Emergency in Mexico, from the National Dataset

For the second part of the study, we conducted a retrospective analysis of the 38 months of the Mexican COVID-19 sanitary emergency, using a start date of 16 February 2020 and an end date of 9 May 2023 (declaration of end of emergency). We used the updated version of the public Mexican COVID-19 data from the General Direction of Epidemiology, available at [7] and extensively described in [3], that recorded all the (symptomatic) COVID-19 cases officially identified in Mexico, since the beginning of the epidemic and continues to be updated weekly. Final dates for the datasets analyzed were 11 October 2022 for information on 2020–2021 and 9 May 2023 (end of sanitary emergency declared) for information on 2022–2023. Data were managed and aggregated in R/R studio, according to the flowchart in Figure 1. Appendix A includes the total number of cases and deaths analyzed per wave, which corresponds to all the national cases in adults available from the beginning to the end of the sanitary emergency. Percentages, odd ratios (OR) and relative risk (RR) with 95% confidence intervals were determined in GraphPad Prism 7. Chi-square tests with Yates continuity correction and Fisher’s exact tests were performed in GraphPad Prism 7. The Case Fatality rates (CFR) corresponded to the percent of detected cases that died. The computer code used in the analysis is available at DOI: 10.5281/zenodo.8397772. The original code was produced and released with [21] but has been since updated. The code version used for this analysis was the modification released by us on 2 October 2023.

## 3. Results

### 3.1. Factors Associated with Severe COVID-19, before Vaccination in Mexico, from the Published Literature

#### 3.1.1. COVID-19 Hospitalization, Pneumonia, AMV and ICU

In publications prior to SARS-CoV-2 vaccination, detected pneumonia posed the highest risk for hospitalization (OR = 33) as it marked a state of severe disease (Table 1), while the comorbidities most associated with COVID-19 pneumonia or hospitalization, were CKD and IS alone or in combination with diabetes, hypertension or obesity (OR > 2) (Table 1 and Appendix A) [13,14,15,16,17,18,19,20]. Patients aged > 50 yo had higher odds of hospitalization (OR > 2.0) (Table 1). Obesity, diabetes, or hypertension, alone or combined with other comorbidities, were important risk factors for AMV (OR > 1.4), ICU (OR > 1.3), or hospitalization (OR > 1.3) [8,13,14,15,16,17,18,19,20]. Mejia-Vilet [11] proposed nine parameters, that predicted up to 99% of ICU hospital admission: male gender, obesity (Body Mass Index, BMI ≥ 30 kg/m^2^), systolic blood pressure < 100 mmHg, Charlson index ≥ 3, glucose > 200 mg/dL, albumin < 3.5 mg/dL, lactate dehydrogenase (LDH) > 474 U/L, SaO_2_/FiO_2_ ratio < 300, and lung damage > 50%.

#### 3.1.2. Factors Associated with COVID-19 Death

In the analyses published before vaccination in Mexico, the risk of COVID-19 death increased significantly with age (Table 2) [21]. The requirement for hospitalization or AMV was associated with death, with OR > 5.0 [13,18], as it marked a state of severity that required intervention and ranked similar to the risk posed by presenting COVID-19 at age > 60 yo (Table 2). Males showed a greater risk of death than females at every age [13,18,19,21]. The same comorbidities as in Table 1, CKD, IS, diabetes, and combinations with obesity and/or hypertension, were associated with COVID-19 death (Table 2), with risk values strongly dependent on age (the risk added by comorbidity decreased with age) [21]. Obesity and/or diabetes, increased the risk of death in particular at ages 20–39 and 40–59 yo [21]. In contrast, in pediatric patients, the main risk factor for death was pneumonia [22] (Table 2).

Mancilla-Galindo [12], with a multivariate model for predicting COVID-19 death, reported a higher risk of death for adults older than 40. Soto-Mota [10], adjusted and validated the Low Harm Score, to predict mortality in hospitalized patients, with the tests available at hospitals without access to inflammatory biomarkers, using: lymphopenia < 800 cells/µL, oxygen saturation SpO_2_ < 88%, leukocytes > 10,000 cells/µL, hypertension, serum creatinine > 1.5 mg/mL, cardiac damage with creatine phosphokinase (CPK) > 185 U/L and elevated troponin. Vidal [23] reported that elevated alanine transaminase (ALT) > 61 U/L, C-reactive protein > 231 mg/L, and LDH > 561 U/L were associated with death in the hospitalized. Rizo-Tellez [24] reported the IL-15/albumin ratio as a predictor of mortality with a cut-off value > 105.4 in hospitalized COVID-19 cases.

### 3.2. Risk of COVID-19 Death in Cases with Comorbidity by Age Group, before, during and after Vaccine Introduction

From the first COVID-19 waves worldwide, it was clear that comorbidity increased the risk of fatality, but with different risk ratios across ages. In the Mexican literature before vaccination, the COVID-19 case fatality rate (CFR) in adults of all ages without comorbidity was 5.2% (95% CI 3.8–6.6) vs. 16.6% (95% CI 13.8–19.4) with comorbidity [13,16,21,25]. We found similar values when compounding the raw data from waves 1 and 2 (before vaccination): CFR 5.1% vs. 17.8% in adults > 20 yo, without- and with-comorbidity (Table 3), that decreased when considering the entire sanitary emergency to 2.0% (without comorbidity) and 11.0% (with comorbidity) (Table 3, entire emergency, all adults).

The overall CFR in adults during the health emergency in Mexico amounted to 4.8% (Table 3), taking into account n = 331,436 deaths and 6,933,618 cases (Appendix A, last column). These CFRs are overestimates as not all cases were detected or recorded in the official surveillance system, which only included symptomatic individuals. Symptomatic infection detection levels also varied. In particular, in the first wave, all detection depended on RT-PCR, before the introduction of antigen tests, and likely had lower detection of mild cases than other waves.

There are few analyses on how the risk of COVID-19 death with comorbidity behaved in Mexico since vaccines were introduced (ref [2] mainly explores diabetes). To further explore this, we conducted analyses per wave, as each had unique characteristics determined by the circulating variants and the increasing population immunity. Waves after the SARS-CoV-2 vaccine introduction had progressively fewer deaths, despite persisting viral circulation (Figure 2a,b). CFR decreased over time as population immunity grew (Figure 2c, Table 3). During waves 5 and 6 after vaccination, there was also a down-trend in detected cases in the surveillance system (Figure 2a,b), perhaps associated with less viral circulation, milder cases and fewer test-seeking behaviors after vaccination. Still, low lethality was observed in the cases detected during the fifth and sixth waves (Table 3, Figure 2c).

In all waves, CFR increased markedly with age, both with and without comorbidity (Table 3, Figure 2c). In each wave and age group, individuals with comorbidity had higher CFR than their counterparts without comorbidity, with OR > 4.6 in 20–39 yo, which dropped progressively with age, tending to OR values below 1.8 in elders 80+ (Table 2). Thus, comorbidity increased the odds of death more in the young age groups, where the healthy counterparts have very low COVID-19 case fatality.

The OR from comorbidity increased progressively in waves 3 and 4 (during vaccination) (Table 3), suggesting that vaccination with the primary series was more effective in healthier individuals (without comorbidity) at each age, as expected from their more robust immune systems. However, in the fifth and sixth waves, after booster doses were offered to all adults, the young (20–39) and middle-aged (40–59) decreased their OR from comorbidity, whereas older age groups retained similar or slightly higher ORs than in wave 4 (Table 3). This suggests that boosters prevented severe COVID-19 more in young adults with comorbidities than in older adults with comorbidities.

### 3.3. Gender Differences in COVID-19 Deaths, before and after Vaccination

The adult Mexican population contains an estimated 52% females and 48% males. In adults, during the sanitary emergency, 46.1% of COVID-19 cases and 61.5% of deaths were in males. This bias towards more male deaths persisted after vaccination in all adult age groups, furthermore, it was more prominent in populations without comorbidities where overall 67.7% of COVID-19 deaths were in males, with a slight decrease after vaccine introduction (Figure 3a). When considering COVID-19 deaths, females had more comorbidities than males; in particular, over half of the females 60–79 yo that died reported 2+ comorbidities vs. only 40% of males, suggesting that females needed more comorbidities to attain the risk levels of males for severe COVID-19. The types of comorbidities reported in adults who died from COVID-19 were similar between genders, except females tended to have more diabetes while males had a higher frequency of smoking.

In contrast to adults, in children 0–19 yo, the 1730, COVID-19 deaths registered during the emergency were more balanced between genders with 52.0% in males and 48.0% in females. This proportion remained similar even in the youngest, 0–4 yo (n = 753 deaths) that didn’t have access to COVID-19 vaccines during the emergency, with 52.6% of COVID-19 deaths in males (not shown).

### 3.4. Frequency and Age Distribution of COVID-19 Cases and Deaths with and without Comorbidity in Mexico, before and after Vaccine Introduction

Before SARS-CoV-2 vaccination in Mexico, when mostly severe cases were detected, publications reported that around 46% (95% CI 48.3–43.7) of COVID-19 cases occurred in individuals with comorbidity [13,14,15,16,20,21,25]. As more detection became available adult cases with comorbidity decreased, then stabilized around 25% in waves 3–6, retaining an age gradient with less comorbidity in younger adults (Figure 3b), consistent with the health patterns of the general population in which chronic disease prevalence increases with age (6). The frequency of comorbidity in COVID-19 deaths has varied less than in cases, with around 70% of deaths reporting at least one comorbidity in every wave so far and without obvious variations with vaccine introduction (Figure 3b). Around a third of the adults that died of COVID-19 in the sanitary emergency, had 2 or more comorbidities, increasing with age to >40% after age 60 (Figure 3d).

When separating adults into 20-year groups, ages 60–79 registered the most deaths in all waves (Figure 3c). After vaccine introduction, there was an important transition towards older deaths, with individuals 80+ gradually increasing from 10% of all deaths in the first wave to >35% in the fifth and sixth waves, in particular at the expense of decreasing deaths in middle-aged 40–59 yo, both in population with and without comorbidity (Figure 3c), again suggesting that vaccines were more effective in the middle-aged than in the elderly.

### 3.5. Frequency of Specific Comorbidities in COVID-19 Deaths in Mexico, before and after Vaccination

The Mexican COVID-19 dataset interrogates the presence/absence of nine preexisting comorbidities in every patient (obesity = OB, diabetes = DM, hypertension = HT, cardiovascular disease = CVD, chronic kidney disease = CKD, chronic obstructive pulmonary disease = COPD, immunosuppression = IS, asthma, smoking, and the option “other comorbidity/ies” which applies to comorbidities not included in the previous 9 categories and may include cancer, liver disease, neurologic and mental disease, etc.). Over 99% of cases in the Mexican COVID-19 dataset contain information for this matrix of comorbidities, so it is known for over 6.8 million COVID-19 cases and 329,520 COVID-19 deaths during the emergency. Hypertension and diabetes were the most frequent comorbidities in COVID-19 deaths in every wave (Figure 4), present in 45% and 37% of COVID-19 deaths, either alone or in combination with other comorbidities. This is consistent with the fact that >90% of deaths have been at ages 40+, a population that accumulates a high prevalence of these diseases (over 30% of adults 40+ are hypertense and around 20% are diabetic) (6).

Obesity alone or in combination with other comorbidities, was the third most frequent comorbidity, present in over 20% of COVID-19 deaths in the first three waves, but it markedly decreased its frequency in waves 4–6, after all adults accessed vaccination and boosters (Figure 4). In contrast, less frequent diseases like CKD, COPD, CVD and IS increased their relative frequency in COVID-19 deaths in waves after vaccine introduction (Figure 4), suggesting that vaccines were more effective in younger people with obesity than in the population with severe comorbidities that impair organ reserve. In fact, chronic kidney disease (CKD) surpassed obesity as a comorbidity in COVID-19 deaths in waves 5–6 (Figure 4).

In Figure 5 we show the specific comorbidity combinations found in COVID–19 deaths before (wave 1–2) and after (wave 5–6) vaccination. Notice that the largest bar corresponds to deaths without comorbidity which were stable at around 29% in adults before and after vaccination and more frequent than any specific comorbidity combination, amounting to almost 100,000 adult deaths during the emergency (Figure 5a,b, first bar). Next was the combination of diabetes + hypertension (DM + HT), present in 11% of deaths (37,452 deaths during the emergency), with little change in its relative frequency before and after vaccination (Figure 5a vs. Figure 5b), followed by hypertension and diabetes as single comorbidities, that decreased their relative frequency after vaccination.

Obesity as a single comorbidity was the fourth most frequent specific comorbidity before vaccination, present in 5.4% of adult deaths, but after vaccination, it decreased to place number 14 and was present in only 1% of COVID-19 deaths (Figure 5a,b, pink bar). Comorbidity combinations that included obesity also decreased their relative frequency in COVID-19 adult deaths after vaccination, including the triad of OB + DM + HT, OB + HT and DM + OB, all of which more than halved their frequency in waves 5–6 compared to waves 1–2. In turn, comorbidity combinations with CKD (CKD + DM + HT and CKD + HT), increased their relative frequency after vaccination, as did “other comorbidities”, “smoker-only”, COPD alone or in combination with “other comorbidities”, IS as a single comorbidity and comorbidity combinations with CVD (Figure 5a vs. Figure 5b).

The preexisting comorbidity combinations seen in children 0–19 yo who died from COVID-19 (n = 1730 during the emergency) were different from those of adults (Figure 5 vs. Figure 6). More than half of the pediatric deaths (54%, n = 934) had no comorbidities, followed by the option “other comorbidity” present in 10.7% of deaths (Figure 6). Then, single comorbidities dominated, like obesity (5.5%), IS (4.2%), CVD (3.3%), CKD (2.7%) or diabetes (2.3%) (Figure 6). In the COVID-19 surveillance system, diabetes is not separated into type 1 and type 2, but likely, preexisting type 1 diabetes was more frequent than type 2 in the pediatric group, as it is 2 to 3 times more prevalent than type 2 in the pediatric population [32]. Preexisting hypertension alone, was present in only 1.3% of pediatric COVID-19 deaths (Figure 6), with a frequency 10 times lower than in adult deaths (Figure 5 vs. Figure 6).

### 3.6. Mean Age at COVID-19 Death in Adults with and without Comorbidities

In Mexico, the mean age at COVID-19 death differed by comorbidity, following the trends present in the general population, with higher ages for diseases that are prevalent in the elderly (COPD, CVD, HT and DM-HT) [33]. The age at COVID-19 death increased significantly in waves 4 and 5–6, with respect to waves 1–2 and 3 for all the specific comorbidity combinations explored, except IS and the option “other comorbidity” (Figure 7, Appendix A). In adults without comorbidity, the age of COVID-19 death after vaccine roll-outs increased by a mean of 7.8 years; while in single comorbidities, and in the most frequent comorbidity combinations (all combinations between obesity, diabetes, hypertension and CKD) the mean age increased from 5 to 10 years (Figure 7 and Appendix A). The largest age increments were found for OB + HT (10 years), CVD as a single comorbidity (9.7 years) and hypertension as a single comorbidity (9.2 years), suggesting that young and middle-aged individuals with these comorbidities benefited a lot from vaccination/hybrid immunity. In turn, the smallest age increments after vaccine roll-outs were found for IS (mean age decreased by 3 years, not significant), “other comorbidity(ies)” (2 years, not significant), HT + CKD (5.3 years), and DM + HT + CKD (5.35 years). Obesity as a single comorbidity had a mean increase in age at COVID-19 death of 6.5 years (Appendix A). The increase in mean age at COVID-19 death detected after vaccine introduction suggests that vaccines protected young and middle-aged individuals, both with and without comorbidity, as shown in Figure 3c, although Figure 7 and Appendix A show that the effect size depends on the comorbidity combination analyzed. No increment in age was observed after vaccine roll-outs for individuals with IS whose impaired immune systems benefit less from vaccines.

### 3.7. Relative Risk for COVID-19 Death in Adults, Derived from Specific Comorbidity Combinations, before, during and after Vaccination in Mexico

Next, we calculated the relative risk (RR) of death per wave and age group, for the 17 most frequent comorbidity combinations, in individuals who presented COVID-19 compared to their counterparts of the same age group without comorbidity. Figure 8 graphs the RRs, to compare their magnitudes across ages and waves. Figure A2 is a regraph of the RRs, separated by age group, into young (20–39 yo), middle-aged (40–59 yo), and senior adults in 60–79 yo and 80+ yo groups. Notably, CKD as a single comorbidity and its combinations with DM, HT or DM + HT posed the highest risks, followed by IS (Figure 8 and Figure A2). The combinations of DM with HT and/or OB also ranked high in risk of COVID-19 death, followed by CVD as a single comorbidity, “other comorbidities”, COPD and diabetes alone; while hypertension, obesity and their combination ranked slightly lower in risk. In turn, being a smoker only provided risk in the older age groups and asthma was not a significant risk factor for COVID-19 death at any adult age or wave (Figure 8 and Figure A2).

In Figure 8 it can be appreciated that the relative risk from comorbidities decreased markedly with age in all waves and comorbidity combinations assayed, except for asthma and smoking. This is because all the RRs were calculated comparing the adverse outcomes of people of the same age group with- vs. without comorbidity, and seniors 60+ have high fatality rates even in the absence of comorbidity. The increased risk at young ages is accentuated during waves 4 and 5–6 (after vaccination) for severe comorbidities like CKD and its combinations, IS and “other comorbidity” (Figure 8). In contrast, obesity as a single comorbidity had a markedly different behavior and was the only comorbidity that decreased its risk after vaccination, becoming a non-significant risk factor in waves 5–6 (Figure 8).

Relative Risk behavior across waves shows age differences (Figure A2 in Appendix B). For the young adults 20–39 yo, all the explored comorbidities remained similar in waves 4 vs. 5–6, with a tendency to decrease, in particular for obesity and diabetes as single comorbidities, suggesting persistent protection from vaccines/hybrid immunity in this age group. The middle-aged adults had a clear RR decrease in waves 5–6 (after vaccination and boosters) in most comorbidities, in particular, the most prevalent like obesity, hypertension, diabetes and their combinations. In turn, COPD and “other” comorbidities showed no change in RR in the middle-aged adults as vaccination/hybrid immunity increased, while those comorbidities with higher risk, like IS, and CKD, actually increased their RR importantly in wave 4 and then again in waves 5–6, suggesting that protection from vaccines + boosters was better for the middle-aged without comorbidities than in individuals with preexisting conditions. Something similar happened in seniors 60–79, but not in the 80+, whose risks remained closer to the threshold of 1 (no risk) across waves (likely because, as discussed, they have high fatality even in the absence of comorbidity). Only “other” comorbidities and COPD, tended to increase the 80+’s risk in waves 5–6 vs. 4.

### 3.8. Risk of Death in COVID-19 Cases without Comorbidity by Age Group, before and after Vaccination in Mexico

To further characterize the population protection provided by vaccination coupled with growing hybrid immunity, we analyzed the Relative Risk of COVID-19 death, in adults without comorbidity, that is, excluding the diverse group of individuals who had worse prognoses due to the various comorbidities analyzed above. Waves 1–2 were used as the point of comparison to represent the situation before vaccination. Each wave since vaccine introduction showed significant (*p* < 0.001) and increasing protection against COVID-19 death in all age groups (RR < 1) (Figure 9). In wave 3, caused by the Delta variant, protection was modest for the 20–39 yo (95%CI RR 0.8–0.87) and the 40–59 yo (95%CI RR 0.53–0.56), who just started accessing vaccines as wave 3 evolved. The elderly 60+ had access to complete primary series, without boosters, right before wave 3 [1], and displayed RR values of 0.52–0.55 for ages 60–79 yo and 0.66–0.71 for 80+ in wave 3 (Figure 9).

In wave 4, when all adults had accessed the complete primary series and the 60+ accessed the first boosters, all adults without comorbidities had a substantial increase in protection (Figure 9). Wave 4 was produced by ample circulation of the first Omicron variants and resulted in many infections that further increased hybrid immunity, and at the end of that wave vaccine boosters were again offered to all adults (many received fourth doses). These factors likely contributed to another increase in protection seen in wave 5 in all adult groups without comorbidity, which persisted through wave 6 (Figure 9), despite no new boosters massively applied for over a year.

In this comparison using only individuals with no reported comorbidities (Figure 9), the age group 40–59 yo showed the most protection, and the elderly 80+ showed the least, in waves 4 through 6. In turn, in waves 5–6, after vaccination and boosters, the elderly 60–79 yo had similar protection to the young 20–39 yo, likely because two boosters were offered to the 60+ whereas younger adults were eligible for only one booster and had somewhat lower vaccination coverage by April 2022 (Figure A1).

### 3.9. COVID-19 Deaths in the Middle-Aged in Mexico, before, during and after Vaccination

The age for COVID-19 death during the pandemic has been lower in Mexico and Latin America than in higher-income countries (Figure 10a), in part related to a younger population pyramid in Mexico where only 3% of the population is age 75 or older and 0.8% are 85 or older (about one million Mexicans are 85+) (Figure 11c, table inset). Positive Pearson correlations (>0.85) suggest that demographics are responsible for at least part of the COVID-19 death distribution among young and senior ages across countries (Figure 10b).

In Mexico, before vaccination (waves 1–2) 50.5% of COVID-19 deaths were in individuals younger than 65 yo (Figure 10 and Figure 11a) and still increased to 56.1% in the first wave after vaccine introduction (wave 3, Delta variant) when only population aged 60+ had accessed vaccination [1,3]. In contrast, in most high-income countries, only around 15% of COVID-19 deaths happened in individuals younger than 60–65 yo even pre-vaccination (Figure 10). The trend of young COVID-19 deaths in Mexico began reverting in wave 4 after all adults were eligible for vaccination and the roll-out of boosters began, with a notorious shrinking of middle-aged deaths (40–49 and 50–64 yo) and expansion of the proportion of deaths at ages 75+ (Figure 11a). During waves 5 and 6, after vaccination, less than a third of COVID-19 deaths in Mexico were in individuals younger than 65 yo (Figure 11a), approaching the trends present in the USA, but higher than observed in the UK, that had less than 16% of deaths under 65 yo in all the pandemic years (Figure 11b). Both Mexico and the USA experienced a decrease in the proportion of middle-aged deaths (40–49 and 50–64 yo) towards the end of the sanitary emergency, as vaccination and hybrid immunity increased. Mexico has had a larger proportion of COVID-19 deaths at young (0–39 yo) and middle-ages (40–64 yo) than the USA and higher overall and per-year mortality rates (MR), except in the elderly and in 2022, when the MR decreased substantially in Mexico post-vaccination (Figure 11b,c). The USA and The UK have similar population pyramids (Table in Figure 11) yet the USA exhibited larger proportions of COVID-19 deaths in middle aged adults, highlighting that the phenomenon is complex, not driven solely by the countries’ demigraphics and may have components related to population health, access to health services, etc.

## 4. Discussion

Mexico is a relevant scenario to explore the interplay of age, chronic comorbidities and vaccination on the risk of severe COVID-19, mainly because: (1) health authorities maintain a national dataset [7], collected whenever a patient seeks medical attention for mild or severe respiratory disease, that records COVID-19 test results and outcomes prospectively along with a matrix of comorbidities; (2) Mexico was one of the countries with more deaths in middle-aged and young adults during the sanitary emergency (Figure 10a and Figure 11), which along with the high prevalence of metabolic disease allow analyses with significant numbers across ages; (3) Mexico implemented a massive SARS-CoV-2 vaccination strategy, with defined periods per age group; thus, periods before, during and after vaccination can be distinguished.

The main limitation of our study is that the national COVID-19 dataset doesn’t record whether each case was vaccinated, so our study did not conduct a direct analysis of vaccine effectiveness, but rather compared population case fatality and mortality trends in periods with/without vaccination. Yet this conveys a general real-world picture of the population effect of vaccination in Mexico. Another limitation is that vaccines from seven developers were used in Mexico, each with different efficiencies that have been documented in the Mexican population [2], thus not all vaccinations had the same quantitative effect. Also, a limitation is that the national COVID-19 dataset was created for surveillance purposes, without a research design in mind. Due to the widespread national effort of information collection, there may be differences in the depth with which comorbidities were interrogated. Obesity was recorded but not overweight which is also highly prevalent in the adult population and individuals with undiagnosed comorbidities were recorded as “without comorbidity”.

Mexican adults had good acceptance of SARS-CoV-2 vaccination [1] and about 85% received at least one vaccine dose. Simultaneously, 95% had anti-N SARS-CoV-2 antibodies [1], suggesting high levels of hybrid immunity by waves 5 and 6 (post-vaccination).

COVID-19 vaccination coverage in Mexico is similar to the world [36] and OECD [35] averages, and about 7% lower than that reported by the USA [37] where 92% of adults 18+ had received at least one SARS-CoV-2 vaccine dose by May 2023, vs. 85% in Mexico [1]. In the USA, 79% of adults 18+ completed a primary series vs. around 72% in Mexico [1], while in the USA 20% had received an updated booster by May 2023, while in Mexico updated boosters were not introduced in 2023. Coverage differences are larger in the pediatric population since only 20% of children 5–11 yo and 40% of 12–17 yo have received a two-dose vaccine series in Mexico [1] while in the USA primary series coverage surpasses 50% at these ages [37].

Vaccination in Mexico differed from that of higher income countries because only 20–30% of adult vaccine doses were with mRNA vaccines, most primary series used two doses of the same vaccine, but most boosters were heterologous; no updated boosters have been introduced and more than a year has passed since the massive application of the last boosters. In this context, we asked whether the hybrid population immunity attained by early 2022 was still protective against COVID-19 death a year later, when mostly Omicron variants continued to circulate, that could evade immunity from vaccination and infections with ancestral/previous strains [38]. We used the above-mentioned national COVID-19 dataset, to analyze epidemic dynamics by waves, and compared periods before, during and post-vaccination. Health authorities continue to update the national COVID-19 dataset serving as an important resource for retrospective analyses like ours.

Our main findings indicate that, as expected, COVID-19 mortality (Figure 11) and case fatality decreased importantly after vaccination in Mexico (Figure 2). Better treatment protocols, less hospital saturation and milder virus variants may also have contributed to the decrease in severe COVID-19 during 2022 and 2023. In 2020–2021 COVID-19 was the second most frequent cause of death in Mexico, just narrowly surpassed by cardiovascular deaths and resulted in excess mortality above 40% particularly affecting middle-aged adults [39]. In contrast, in 2022–2023, after all adults accessed vaccines, COVID-19 disappeared from the first five causes of death, which went back to pre-pandemic trends (cardiovascular disease, diabetes mellitus, cancer, liver disease and accidents). This was accompanied by the cessation of excess mortality early in 2022 [3] by the end of wave 4, produced by Omicron BA.1 variants. Along with the decrease in COVID-19 mortality, we detected several important trends that changed with population access to vaccines, and are reported here:
Mean ages at COVID-19 death increased with vaccination, both in sub-populations with and without comorbidities (Figure 7). This effect required that all adults accessed vaccination and was not present during wave 3 (produced by the Delta variant) when only the 60+ had accessed vaccines. In wave 3, the age of COVID-19 death actually decreased for most comorbidities, in particular obesity (Figure 7) and the proportion of young deaths grew, reaching its maximum in the epidemic (Figure 11a). This highlights the importance of vaccine access to all adults, even the youngest for epidemic control.Obesity stopped being an important risk factor for COVID-19 death after all adult age groups were eligible for vaccination and boosters (Figure 4, Figure 5 and Figure 8).Population access to vaccines/hybrid immunity protected all adult age groups from severe disease including the elderly (Figure 9). This protection was lower in the 80+ than for other adult groups but was maintained despite no new boosters applied in the year previous to this analysis. Lethality didn’t increase during the sixth wave (Figure 2c), despite a longer time since the last boosters, again suggesting that the protection from vaccines/hybrid immunity persisted through this wave.Middle-aged adults 40–65 exhibit the most protection from vaccines/hybrid immunity (Figure 9) along with a pronounced contraction of deaths at these ages (Figure 3c and Figure 11a,b).

It is well known that COVID-19 displays more severity at older ages, attributed to the gradual decline of innate and adaptive immunity (immunosenescence) [40,41,42], which limits the response against infections and the response to vaccines [43]. The COVID-19 pandemic illustrated that this severity spectrum is also influenced by the general health of individuals and shifts towards severity if comorbidities are present, in particular severe comorbidities and multimorbidity that lower organ reserve, impair the immune response/emergency hematopoiesis [40] and serve as general markers of aging. This gradient is illustrated in the current population analysis where, despite the general tendency towards lower case fatality as population immunity grew, two important trends were retained in every wave: (1) CFR increased importantly with age and (2) CFR remained higher for the population with vs. without comorbidities (Figure 2c).

In Mexico, highly prevalent metabolic comorbidities like obesity, diabetes and hypertension were important risk factors for severe COVID-19 before vaccination, especially in combination with each other, providing more risk at younger ages where the healthy counterparts had much lower CFR [8,13,14,15,16]. This changed with the vaccine introduction which notoriously decreased the risk from obesity. In contrast, risk from diabetes or its combinations persisted after vaccines suggesting that diabetes impairs the immune response associated with vaccines and hybrid immunity more than obesity. Diabetes has been distinguished as an important and persisting risk factor for severe COVID-19 before and after vaccine roll-outs in different countries [2,44,45,46].

Of the comorbidities recorded in the Mexican dataset, chronic kidney disease (CKD) has posed the greatest risk for severe COVID-19 in Mexico, similar to what has been reported in other countries [25,47] and likely related to the low organ reserve of these patients to resist severe infections. Despite being much less prevalent than metabolic disease, severe diseases like CKD and COPD became more frequent than obesity in COVID-19 adult deaths after vaccination (Figure 4 and Figure 5) and retained a high relative risk of death that even increased after vaccine introduction, suggesting a less robust response to vaccines/hybrid immunity in those patients. Patients with immunosuppression also retained a risk of death after vaccine introduction and were the only comorbidity group that didn’t increase their mean age at COVID-19 death. A direct evaluation of vaccine effectiveness in populations with CKD, COPD or IS in Latin America, is lacking.

Finally, our study agrees with the worldwide notion that COVID-19 vaccination has been highly effective in decreasing COVID-19 mortality and that Mexico had a successful control of the COVID-19 epidemic largely aided by vaccines. However, in low and middle-income countries like Mexico, the younger demographics require that public health strategies focus on young and middle-aged adults. Middle-aged adults in Mexico have increased risk both because of incipient immunosenescence and because they carry a high prevalence of metabolic disease from an early age, and they may not behave as their counterparts from higher-income economies (Figure 10 and Figure 11). It is likely that populations in middle- and low-income countries age faster due to harsher living conditions, poorer diets, less time for self-care and exercise and more exposure to pollution, but this effect has not been directly measured. Studies and interventions are needed to address this and preventive efforts like vaccination should include the middle-aged.

Mexico experienced larger mortality rates from COVID-19 in children than higher-income countries like the USA (Figure 11c), and more than half of the pediatric deaths happened without comorbidities (Figure 5c). Mexico hasn’t completed COVID-19 vaccination in children. Children 0 to 4 years old remain not eligible for SARS-CoV-2 vaccination and children 5 to 17 years old have partial coverage with no access to third doses. The pediatric population did not show excess mortality during the sanitary emergency in Mexico, and during the sanitary emergency, vaccines went first to adult groups exhibiting more fatalities. However, now that the emergency has passed it is important to improve SARS-CoV-2 vaccination coverage for pediatric groups, as the virus will continue to circulate and vaccines are efficient and safe.

## Figures and Tables

**Figure 1 vaccines-11-01676-f001:**
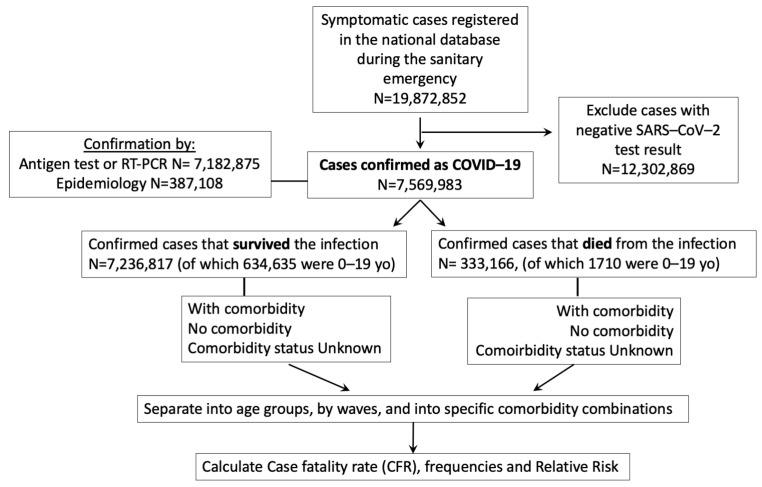
Flowchart of the analysis conducted with the Mexican COVID-19 dataset from the entire sanitary emergency (28 February 2020 to 9 May 2023). Some individuals may be represented more than once if they had more than one symptomatic infection. Asymptomatic infections are not represented in the Mexican COVID-19 dataset.

**Figure 2 vaccines-11-01676-f002:**
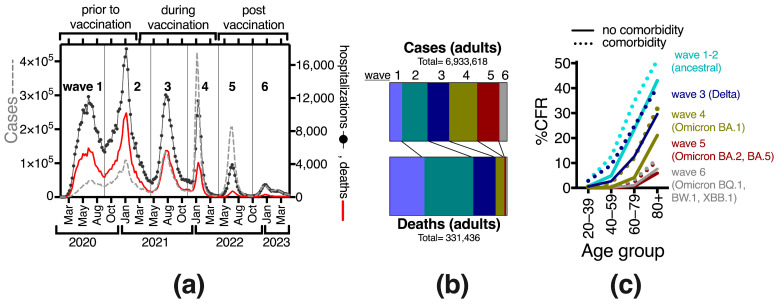
The lethality of COVID-19 gradually decreased in Mexico during the sanitary emergency. (**a**) COVID-19 epidemic curve illustrating the six waves that occurred during the emergency in 2020–2023. Left *y*-axis shows cases, and right *y*-axis hospitalizations and deaths (all ages). (**b**) Proportion of deaths in adults with respect to cases, distributed by wave. (**c**) Case fatality rate (CFR) across waves, by patient’s age. In this panel, comorbidity refers to any single or combination of preexisting conditions, including smoking.

**Figure 3 vaccines-11-01676-f003:**
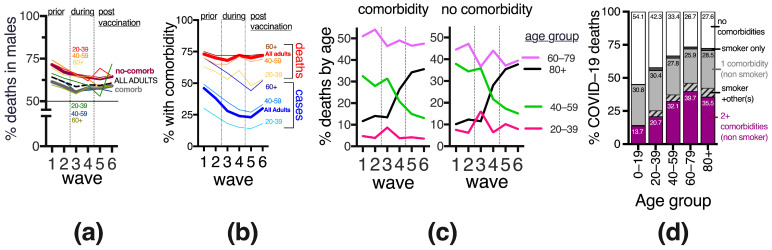
Development of the COVID-19 sanitary emergency in adults in Mexico, across the six waves (2020–2023), and in relation to vaccine introduction, in terms of (**a**) proportion of deaths in adult males, (**b**) COVID-19 cases and deaths with and without comorbidity; (**c**) rate of deaths by age group separated into subgroups with and without comorbidities. (**d**) A summary (not separated by wave) of the COVID-19 deaths by age group and number and type of comorbidities. Numbers inside the bars state the proportion (%) of deaths with that condition; 100% is the total deaths in the age group. “Smoker + other(s)” refers to any combination when the patient was a smoker and reported another preexisting disease.

**Figure 4 vaccines-11-01676-f004:**
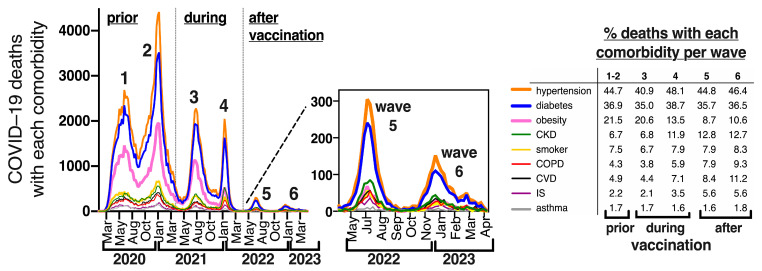
Comorbidities in COVID-19 deaths in Mexico (all ages) represented as epidemic curves through time during the sanitary emergency. Each color represents a comorbidity of the 9 interrogated in the dataset, irrespective of whether it is presented alone or in combination with other conditions. The insert zooms into waves 5 and 6. The table shows the specific percentage of each comorbidity (alone or in combination with others) per wave; 100% was the total number of COVID-19 deaths in each wave. CKD = chronic kidney disease, COPD = chronic obstructive pulmonary disease, CVD = cardiovascular disease, IS = immunesuppression. The numbers to calculate the percentages shown in this insert table can be found in Appendix A.

**Figure 5 vaccines-11-01676-f005:**
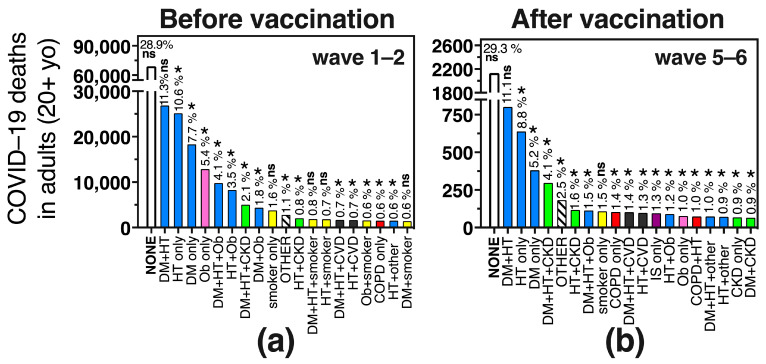
The 20 most frequent comorbidity combinations in adult COVID-19 deaths in Mexico, before (**a**) and after (**b**) SARS-CoV-2 vaccination, that account for 84% and 77.8% of total COVID-19 deaths in the waves shown. DM = diabetes mellitus, HT = hypertension, Ob = obesity, CKD = chronic kidney disease, COPD = chronic obstructive pulmonary disease, CVD = cardiovascular disease, IS = immunesuppression. Combinations between the conditions related to metabolic syndrome (diabetes, hypertension and/or obesity) are in blue bars; combinations with CKD are in green, combinations with being a smoker are in yellow, with CVD in black and with COPD in red. Obesity without other comorbidities is in a pink bar and IS in a purple bar; * indicates that the frequency was different in (**a**) vs. (**b**), with *p* < 0.0001 (Chi2 and Fisher exact test), while ns indicates that the frequency was not significantly different. The comorbidity combinations in adult COVID-19 deaths for the entire duration of the sanitary emergency (not separated by waves) are depicted in Appendix A.

**Figure 6 vaccines-11-01676-f006:**
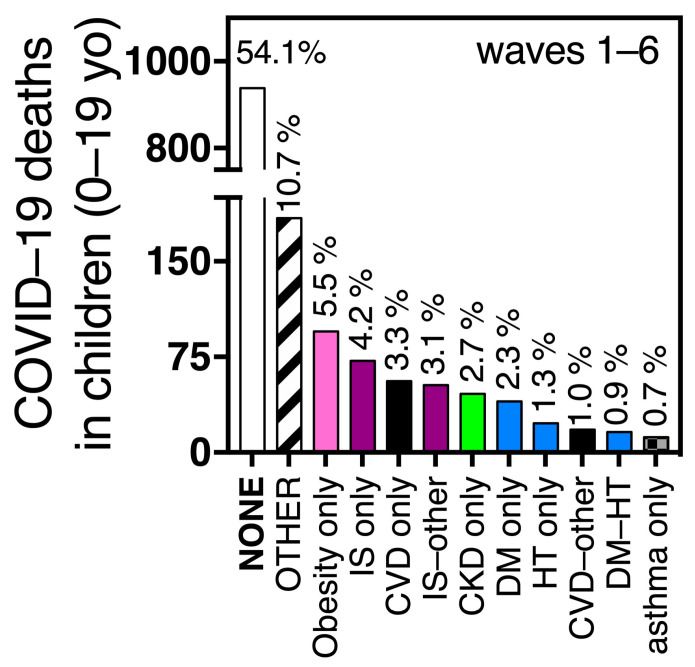
The most frequent comorbidity combinations found in pediatric COVID-19 deaths during the sanitary emergency in Mexico. The combinations depicted here account for 89.8% of pediatric deaths. Abbreviations and color codes are as in Figure 5. Pediatric deaths were not separated by waves in this figure as children accessed vaccination later than adults, mostly during waves 4 and 5, not all ages were eligible, and their coverage is lower than in adults so no clear distinction before and after vaccination can be made.

**Figure 7 vaccines-11-01676-f007:**
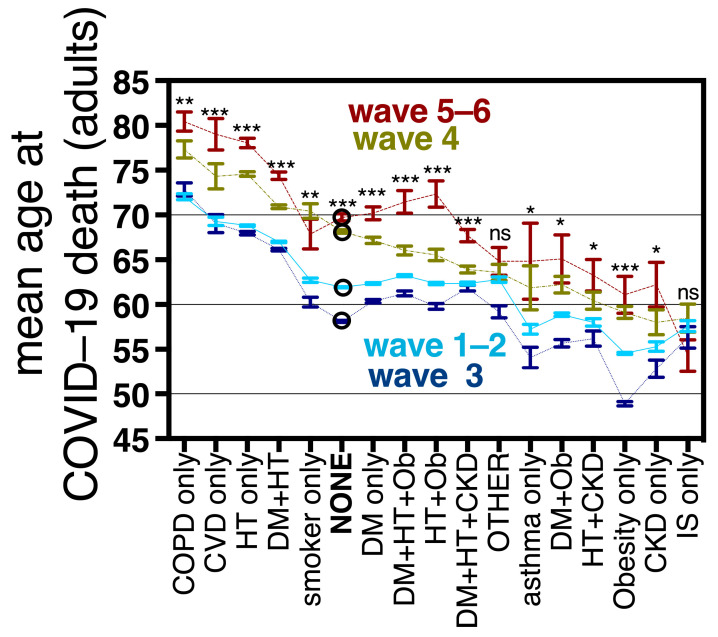
Mean age and standard error at COVID-19 death of the most frequent comorbidity combinations found in the Mexican dataset, by wave, including the option “no comorbidity” (depicted as NONE and marked with an empty circle in the graph). Statistical significance in the difference of the mean age for each point in wave 1–2 vs. 5–6 is expressed with asterisks as follows: *** *p* < 0.0001, ** *p* < 0.001, * *p* < 0.05, ns not significant *p* > 0.05. Waves are depicted using different colors; also wave 1–2 (before vaccination) are strung with a solid line, wave 3 with a dotted line, wave 4 with line and dots and wave 5–6 (after vaccination) with a discontinuous line. Mean ages for each wave as well as the differences between waves are in Appendix A. Abbreviations are as in Figure 5.

**Figure 8 vaccines-11-01676-f008:**
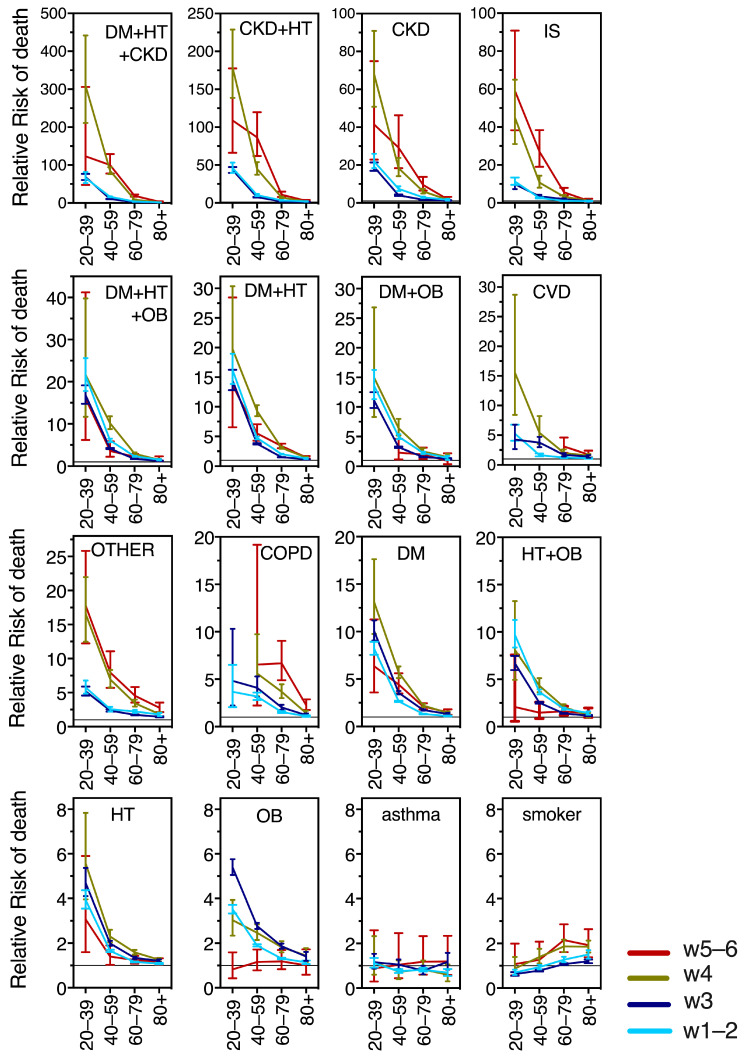
Time course across waves of the relative risk of COVID-19 mortality in adults. The RR data were plotted to compare their magnitudes across the six waves and age groups. Comorbidity abbreviations as in Figure 5.

**Figure 9 vaccines-11-01676-f009:**
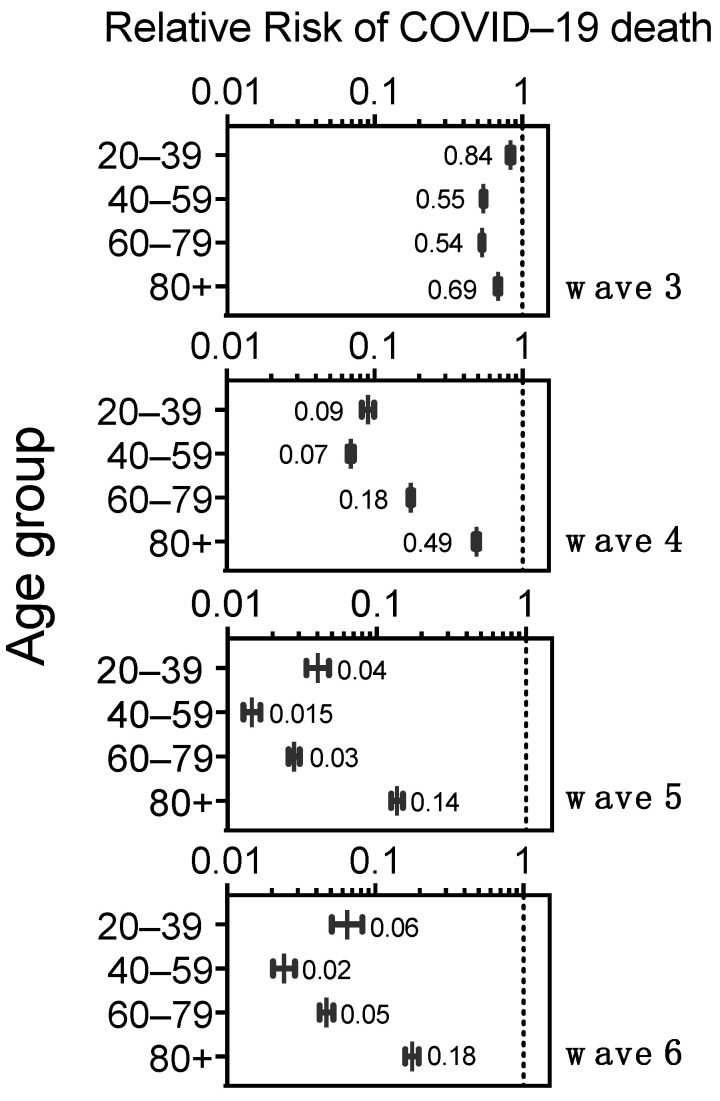
Relative Risk of COVID-19 death in adults without comorbidity, represented in logarithmic scale (base 10) and per age group, to compare each wave after vaccine roll-outs (waves 3–6) against waves 1–2 (pre-vaccination, used as background). All values had *p* < 0.0001 against waves 1–2.

**Figure 10 vaccines-11-01676-f010:**
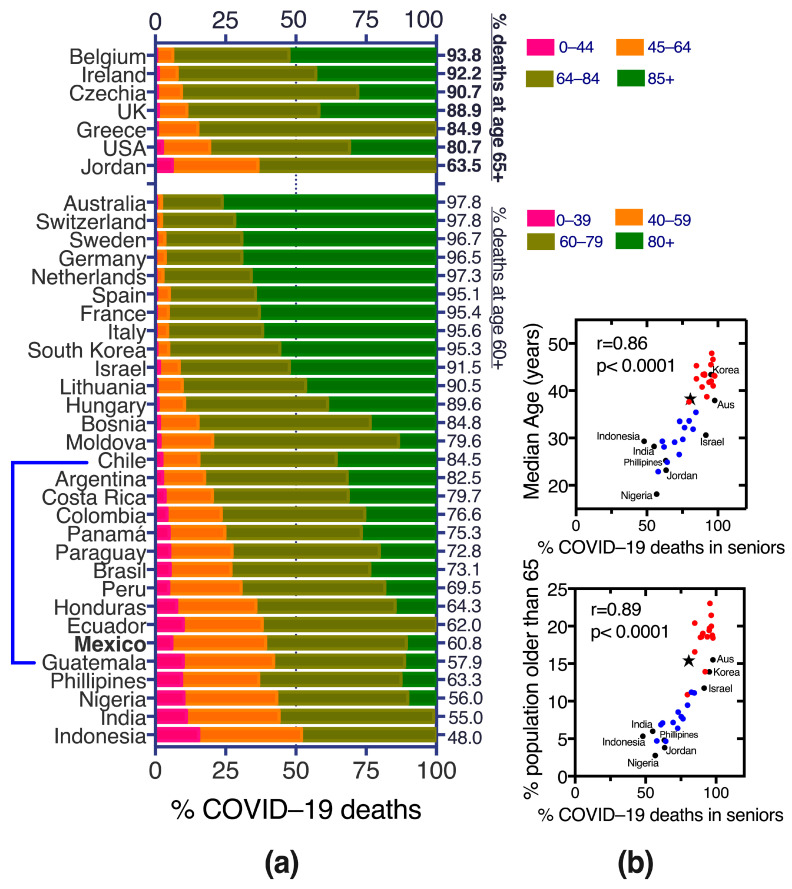
Proportions of COVID-19 deaths per age group and country until early March 2021 (pre-vaccination or with vaccination in progress/incomplete). (**a**) Most COVID-19 deaths in higher income, European countries were in population 60+ or 65+, while Latin America, Philippines, Nigeria, India and Indonesia had a larger proportion of deaths at younger ages. The top part of the graph has a different age partition based on what that group of countries was reporting. Some countries like Indonesia and India, Jordan and Greece did not further separate elderly COVID-19 deaths into 60–79 and 80+, or 65–84 and 85+, thus elder deaths are shown grouped as 60+ or 65+. (**b**) Two variables describing these countries’ demographics (median age of the population and percentage of population older than 65 yo) graphed against the cumulative % of COVID-19 deaths in seniors until early March 2021, had a good Pearson coefficient (r) with *p* < 0.0001. Red dots are European countries and the USA, blue dots are Latin American countries, and black dots are countries from other regions and marked with name. Aus = Australia. Mexico is depicted with a black star.

**Figure 11 vaccines-11-01676-f011:**
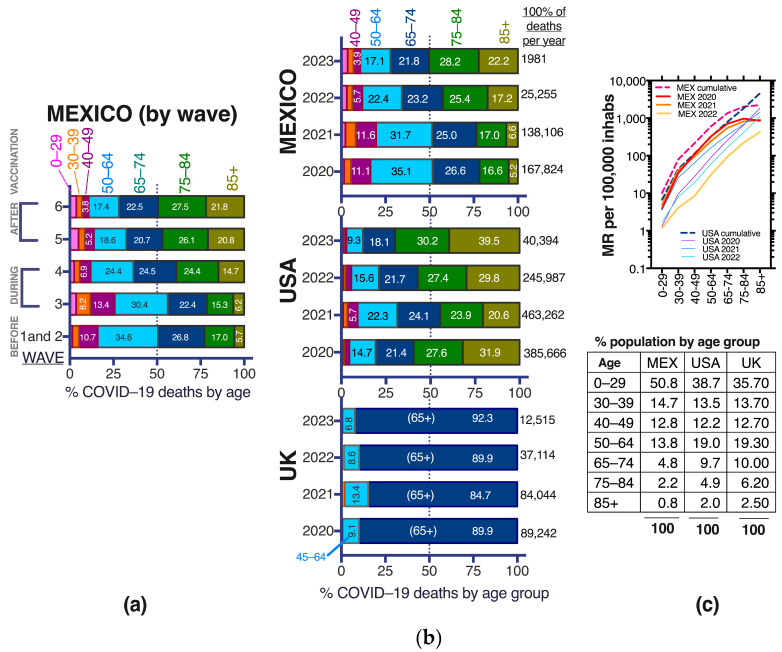
Age demographics of COVID-19 deaths during the sanitary emergency to compare Mexico with other countries. (**a**) Proportion of COVID-19 deaths per age group and wave in Mexico; (**b**) proportion of deaths by age group and pandemic year comparing Mexico, the USA (with data from the CDC [34]) and the UK (as found in OECD stats [35]). Numbers inside the bar sections correspond to percentage of COVID-19 deaths at that age, from the total in the wave or year. The age partitions reported by the USA CDC were used for Mexico and the USA, with the same color code in a and b; while for the UK only three categories are reported in OECD stats: COVID-19 deaths at 0–44 yo (depicted in orange) which stayed below 2% in all the pandemic years, those at 45–64 yo depicted in light blue and staying below 13.4%, and the largest section of the bar corresponding to COVID-19 deaths in elders 65+ (depicted in dark blue) Other OECD (Organization for Economic Development and Cooperation) countries and most of the countries in Figure 10 stopped reporting COVID-19 deaths by age, thus the comparison was limited to these three countries. For 2023 only data until May are included. (**c**) TOP: crude mortality rate (MR) per 100,000 inhabitants (displayed in logarithmic scale base 10) by age group and year and cumulative, in Mexico and the USA. Only whole years were included in MR calculations; BOTTOM: proportion of each population group in Mexico, the USA and the UK to compare the population pyramids, with data from 2021 available at [5].

**Table 1 vaccines-11-01676-t001:** Published risk factors for COVID-19 hospitalization in Mexico, before SARS-CoV-2 vaccination, ranked in decreasing order (with data up to January 2021).

Risk Factor for Hospitalization	Odds Ratio (95% CI)	Reference
Detected pneumonia	33 (29–38)	[13]
CKD + DM	6.6 (2.4–18.3)	[14]
CKD + HT	4.3 (2.0–9.3)	[14]
Age ≥ 75	3.8 (2.9–5.1)	[13]
IS + Ob	4.2 (1.8–9.8)	[15]
	3.7 (1.6–8.8)	
CKD	3.5 (2.6–4.6)	[14]
	2.4 (1.4–4.3)	[13,15,16]
IS	2.9 (1.9–4.6)	[15]
	2.4 (1.9–3.2)	[16]
Age ≥ 65	2.8 (2.7–2.9)	[17]
DM + Ob	2.7 (2.1–3.5)	[14,15]
	1.7 (1.2–2.6)	[13]
DM	2.6 (2.2–3.0)	[14,15]
	2.0 (1.9–2.01)	[13,16,17]
DM + HT	2.5 (2.0–3.0)	[13,14]
Ob + COPD	2.5 (1.1–6.2)	[14]
Age ≥ 50–74	2.0 (1.8–2.3)	[13]
DM and age < 40	2.0 (1.8–2.2)	[17]
Ob + HT	1.9 (1.5–2.3)	[13,14,15]
Ob	1.9 (1.7–2.1)	[15]
	1.6 (1.3–1.9)	[13,14]
	1.4 (1.2–1.6)	[16,17]
COPD	1.9 (1.5–2.4)	[14]
	1.7 (1.2–2.5)	[13,17]
DM + HT+ Ob	1.8 (1.3–2.5)	[13]
Male	1.7 (1.5–1.8)	[17]
	1.5 (1.4–1.7)	[13]
HT	1.5 (1.3–1.9)	[13,14]
	1.3 (1.2–1.6)	[15,16]

CKD = chronic kidney disease; DM = diabetes mellitus; HT = systemic hypertension; Ob = obesity; IS = immune suppression; COPD = chronic obstructive pulmonary disease; CVD = cardiovascular disease. The OR (odds ratio) values of more than one publication are listed together when they were within the 95% CI (confidence interval). Calculations included cases at all ages.

**Table 2 vaccines-11-01676-t002:** Published risk factors for COVID-19 death in Mexico, before SARS-CoV-2 vaccination, ranked in decreasing order (with data up to January 2021).

Risk Factor for COVID-19 Death	OR (95% CI)	Reference
Age > 80	12.5 (10–15)	[18]
AMV	8.7 (3.2–24)	[13]
Age 61–79	7.7 (6.6–9.2)	[18]
	3.7 (2.8–4.9)	[13]
Hospitalization	7.1 (6.8–7.6)	[18]
	5.0 (3.8–6.5)	[13]
Age 41–60	3.7 (2.9–4.6)	[18]
	1.9 (1.6–2.3)	[13]
Pregnancy	3.5 (1.1–10)	[13]
Detected pneumonia	3.4 (3.2–3.5)	[18]
	2.5 (2.1–3.1)	[13]
	6.4 (2–20) ^ped^	[22]
DM + HT + Ob	2.1 (1.5–2.9)	[13]
DM + Ob	2.0 (1.4–3.1)	[13]
CKD	1.8 (1.2–1.3)	[18]
	1.4 (1.1–2.1)	[13]
IS ^HR^	1.8 (1.6–2.0)	[18]
	1.1 (1.0–1.16)	[19]
Ob + HT	1.8 (1.3–2.6)	[13]
Ob	1.7 (1.4–2.2)	[13]
	1.3 (1.15–1.4)	[18]
DM	1.5 (1.13–1.98)	[13,18]
HT	1.5 (1.15–1.92)	[13,18]
Male	1.5 (1.3–1.8)	[13,18]
COPD ^HR^	1.4 (1.3–1.5)	[17]
	1.12 (1.07–1.18)	[19]

Values are Odds Ratios (OR), except the risk of death from IS or COPD, that are HR = Hazard ratios; AMV = assisted mechanical ventilation. The risk values of more than one publication are listed together when they were reported within the same 95% CI (confidence interval). Calculations included cases at all ages, except when marked as pediatric ^ped^ which included only ages < 18 yo. All other abbreviations are as in Table 1.

**Table 3 vaccines-11-01676-t003:** COVID-19 case fatality rate (CFR) and Odds Ratio (OR) of death for adults with comorbidity vs. without, estimated by age group, before, during and after SARS-CoV-2 vaccination in Mexico (national data) during the sanitary emergency.

		Before Vaccination Wave 1–2		During Vaccination Wave 3		During Vaccination Wave 4		After Vaccination Wave 5–6		Entire Emergency	
Age	Comorbidity	% CFR	OR (95% CI)	% CFR	OR (95% CI)	% CFR	OR (95% CI)	% CFR	OR (95% CI)	% CFR	OR (95% CI)
20–39	yes	2.94	4.59 (4.42–4.77) *	2.82	5.27 (5.02–5.52) *	0.50	8.4 (7.45–9.47) *	0.17	5.65 (4.65–6.87) *	1.87	6.06 (5.89–6.23) *
	no	0.66		0.55		0.06		0.03		0.31	
40–59	yes	12.50	2.79 (2.75–2.83) *	9.19	3.68 (3.58–3.79) *	2.03	6.14 (5.78–6.52) *	0.42	5.08 (4.47–5.76) *	7.51	3.92 (3.86–3.97) *
	no	4.87		2.67		0.34		0.08		2.03	
60–79	yes	34.75	1.73 (1.71–1.76) *	24.75	2.28 (2.22–2.35) *	12.11	3.20 (3.06–3.34) *	2.72	3.52 (3.25–3.81) *	24.04	2.44 (2.41–2.47) *
	no	23.52		12.59		4.13		0.79		11.50	
80+	yes	51.13	1.38 (1.34–1.43) *	40.48	1.62 (1.53–1.72) *	31.79	1.75 (1.64–1.86) *	11.32	1.83 (1.68–2.00) *	38.85	1.70 (1.66–1.75) *
	no	43.06		29.58		21.08		6.51		27.17	
All adults	yes	17.82	4.07 (4.03–4.11) *	11.48	5.84 (5.74–5.94) *	4.40	8.44 (8.21–8.68) *	1.30	7.33 (6.97–7.71) *	11.01	5.94 (5.89–5.98) *
	no	5.06		2.17		0.54		0.18		2.04	
All conditions		10.31	-	4.79	-	1.45	-	0.46	-	4.78	-

* With vs. without comorbidity had different CFR with *p* < 0.001.

## Data Availability

R scripts used in this work are available at DOI: 10.5281/zenodo.8397772 Raw data are available at https://www.gob.mx/salud/documentos/datos-abiertos-152127 (deposited by the Mexican government, last accessed on 31 May 2023), and in aggregated form at DOI: 10.6084/m9.figshare.24457312.

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
