# Peer review of "Age and Comorbidities as Risk Factors for Severe COVID-19 in Mexico, before, during and after Massive Vaccination"

_vaccines, 2023, doi:10.3390/vaccines11111676_

Round 1

Reviewer 1 Report

Comments and Suggestions for Authors

This manuscript delineates national data from Mexico about age and comorbidities. The parameters and analyses were sound and straightforward. The outcomes were similar with other countries and continents. I think this manuscript's quality is suitable for publication.

Minor concerns.

1. Suggest adding further discussion about the Omicron prominence waves because the Omicron variant(s) could evade the immunity from vaccination and immunity waning.
The CFR during waves 5-6 remains high in the elderly, even those with booster vaccinations.

Comments.

1. In Table S2 and "2.2", the descriptions were mentioned as "symptomatic".

How about the asymptomatic with detectable RT-PCR, NAAT, or antigen results?
Were they excluded from the database?

2. Line 49. Suggests using "ChAdOx-1 nCoV" or "ChAdOx-1 S" instead of only "ChAdOx".

3. Line 49. Suggests using "University of Oxford—AstraZeneca".

4. Lines 51-52. Suggests using "Pfizer—BioNTech".

5. Lines 153. Suggests subscript "2" in terms SaOâ‚‚/FiOâ‚‚.

Author Response

Minor concerns.

  1. Suggest adding further discussion about the Omicron prominence waves because the Omicron variant(s) could evade the immunity from vaccination and immunity waning.

ANSWER: Thank you for the suggestion. A comment was added in the discussion in fourth paragraph, lines 923-924. 

  1. The CFR during waves 5-6 remains high in the elderly, even those with booster vaccinations

ANSWER: We agree that CFR is high in the Mexican dataset compared to other countries. Our interpretation is that this is related to only symptomatic cases being considered, as stated in the second paragraph of section 3.2, lines 268-272.

Comments.

  1. In Table S2 and "2.2", the descriptions were mentioned as "symptomatic".

How about the asymptomatic with detectable RT-PCR, NAAT, or antigen results? 
Were they excluded from the database?

ANSWER: Asymptomatic individuals were excluded from the data, as the dataset was built only from symptomatic cases that sought medical attention or needed days off from work. This is clarified in materials and methods, line 134 and in the foot of figure 1.

  1. Line 49. Suggests using "ChAdOx-1 nCoV" or "ChAdOx-1 S" instead of only "ChAdOx".

ANSWER: Done, thank you for the suggestion.

  1. Line 49. Suggests using "University of Oxford—AstraZeneca".

ANSWER: Done, thank you for the suggestion.

  1. Lines 51-52. Suggests using "Pfizer—BioNTech".

ANSWER: Done, thank you for the suggestion.

  1. Lines 153. Suggests subscript "2" in terms SaOâ‚‚/FiOâ‚‚.

ANSWER: Done, thank you for the suggestion.

THANK YOU FOR YOUR REVIEW.

Reviewer 2 Report

Comments and Suggestions for Authors

Major comments

1. In general, an international comparison of the findings will be useful, mainly in the discussion but also in other sections.

2. Statistical inference is missing in several analyses, for example: correlations between age and mortality, chi square test to compare vaccination and % of deaths / hospitalizations, the results of figure 5 should also include chi square tests (to compare between the waves, between comorbidities etc).

3. Discussion – a major limitation is the different efficiencies of the different vaccinations. This was not analyzed and it is OK, but a limitation should be clearly written.

Minor comments

1.      Abbreviations section will be useful.

2.      Line 76 – How many Mexicans are of age 85+ ?

3.      Table 1 – please separate it to two sub-tables (hospitalization and death), the lines have no meaning and it is confusing in its current form.

4.      Table 2 – “VAX” is not a common phrase.

5.      Lines 238-240 are more suitable to be in the discussion.

6.      Table within figure 4 – add numbers (not only percentages), this can be in a separate table.

7.      Figure 5c – make it a separate figure.

8.      Line 375- there are two commas (“,”), delete one of them.

9.      Figure 6 – it is hard to understand in black and white.

10.   Figure 6  and line 366 – isn’t COPD more prevalent in older populations? The authors should consider “normalize” each disease to the age in which it occurs.

11.   Consider move Figure 7 to supplementaries. It is too crowded.

12.   Figure 8 can be wider so it will be more readable.

13.   Line 425 “The for the” – please correct.

14.   Figure 9 – why in logarithmic scale?

15.   Line 500 and figure 10 – consider comparing to OECD averages.

16.   Lines 586-587 – please write a reference to support this notion.

Comments on the Quality of English Language

Several small corrections were written to the authors.

Author Response

Major comments

  1. In general, an international comparison of the findings will be useful, mainly in the discussion but also in other sections.

ANSWER: More international context was added, mainly in figure 10, 11 and in the discussion.

  1. Statistical inference is missing in several analyses, for example: correlations between age and mortality, chi square test to compare vaccination and % of deaths / hospitalizations, the results of figure 5 should also include chi square tests (to compare between the waves, between comorbidities etc).

ANSWER: Thank you for the suggestion inference statistics were added to table S2, Figure 5, Figure 7, Figure 9 and Table S5.

3. Discussion – a major limitation is the different efficiencies of the different vaccinations. This was not analyzed and it is OK, but a limitation should be clearly written.

ANSWER: Thank you for the suggestion, this limitation was stated in the discussion, in the second paragraph.

Minor comments

  1. Abbreviations section will be useful: ANSWER: an abbreviation list was added at the end of the manuscript. 
  2. Line 76 – How many Mexicans are of age 85+ ?

ANSWER: There are about a million Mexicans 85+, corresponding to 0.8% of the population. This has been added in section 3.9, lines 741-742.

3. Table 1 – please separate it to two sub-tables (hospitalization and death), the lines have no meaning and it is confusing in its current form.

ANSWER: Done, thanks for the suggestion

4. Table 2 – “VAX” is not a common phrase.

ANSWER: This was changed to vaccination, thanks for the suggestion

5. Lines 238-240 are more suitable to be in the discussion.

ANSWER: Done, thanks for the suggestion. The sentence is now in the discussion, fifth paragraph, lines 933-934.

6. Table within figure 4 – add numbers (not only percentages), this can be in a separate table

ANSWER: Numbers can now be found in Table S4.

7. Figure 5c – make it a separate figure.

ANSWER: Done, former figure 5c is now figure 6.

  1. Line 375- there are two commas (“,”), delete one of them.

ANSWER: Done, thank you.

  1. Figure 6 – it is hard to understand in black and white.

ANSWER: We distinguished each wave by using a different type of line connecting the means.

  1. Figure 6  and line 366 – isn’t COPD more prevalent in older populations? The authors should consider “normalize” each disease to the age in which it occurs.

ANSWER: COPD is indeed more prevalent in older population, we added a reference to support that most COPD related deaths occur in 65+ in Mexico (in section 3.6, line 541). Normalizing comorbidities by age is difficult to conduct along this dataset, since there are no precise statistics of the comorbidity combinations by age group. Comorbidity prevalence data in the general population do not have the same granularity as the COVID-19 dataset. We do state the prevalence of some metabolic chronic conditions in Mexicans (lines 84-86).

  1. Consider move Figure 7 to supplementaries. It is too crowded.

ANSWER: Done, thanks for the suggestion, former figure 7 is now located as annex, as Figure A2, with larger panels

  1. Figure 8 can be wider so it will be more readable.

ANSWER: Done, thanks for the suggestion

  1. Line 425 “The for the” – please correct.

ANSWER: Done, thanks

14. Figure 9 – why in logarithmic scale?

ANSWER: Since the effect spans several orders of magnitude we used a logarithmic scale to facilitate visualization of the different age groups in a single graph, as the effect moves away from the risk threshold=1.

  1. Line 500 and figure 10 – consider comparing to OECD averages.

ANSWER: Thank you for the suggestion. We included figure 10 in the main text (previously was an annex figure) to compare the proportions of young COVID-19 deaths across countries, with data from before vaccination. These omparisons across different countries are dependent on the countries reporting COVID-19 fatalities separated into age-groups and/or complete datasets with age granularity, which became more infrequent as the pandemic weaned. From OECD statistics we found that only the UK is reporting COVID-19 fatalities separated by age groups (using three broad age separations 0-44 yo, 45-64 yo and 65+) and we included this country in the now figure 11.

  1. Lines 586-587 – please write a reference to support this notion.

ANSWER: We added the line “Diabetes has been distinguished as an important and persisting risk factor for severe COVID-19 before and after vaccine roll-outs” and added references.

THANK YOU FOR YOUR REVIEW IT WAS VERY HELPFUL.